biological applications, ecology, environmental science

environmental DNA, species distributions, fisheries, ocean surveys

**Author for correspondence:**
Andrew Olaf Shelton
e-mail: ole.shelton@noaa.gov

# Environmental DNA provides quantitative estimates of Pacific hake abundance and distribution in the open ocean

Andrew Olaf Shelton[1], Ana Ramón-Laca[4], Abigail Wells[2], Julia Clemons[3], Dezhang Chu[3], Blake E. Feist[1], Ryan P. Kelly[5], Sandra L. Parker-Stetter[3,6], Rebecca Thomas[1], Krista M. Nichols[1] and Linda Park[1]

[1]Conservation Biology Division, Northwest Fisheries Science Center[2]Lynker Technologies, Under Contract to Northwest Fisheries Science Center[3]Fisheries Resource Analysis and Monitoring Division, Northwest Fisheries Science Center, National Marine Fisheries Service, National Oceanic and Atmospheric Administration, 2725 Montlake Blvd. E, Seattle, WA 98112, USA
[4]Cooperative Institute for Climate, Ocean, and Ecosystem Studies, University of Washington at Northwest Fisheries Science Center, National Marine Fisheries Service, Seattle, WA 98112, USA
[5]School of Marine and Environmental Affairs, University of Washington, 3707 Brooklyn Ave NE, Seattle, WA 98105, USA
[6]Resource Assessment and Conservation Engineering Division, Alaska Fisheries Science Center, National Marine Fisheries Service, National Oceanic and Atmospheric Administration, 7600 Sand Point Way NE, Seattle, WA 98115, USA

AOS, 0000-0002-8045-6141; AR-L, 0000-0002-9204-6932; BEF, 0000-0001-5215-4878; KMN, 0000-0003-3453-7239

All species inevitably leave genetic traces in their environments, and the resulting environmental DNA (eDNA) reflects the species present in a given habitat. It remains unclear whether eDNA signals can provide quantitative metrics of abundance on which human livelihoods or conservation successes depend. Here, we report the results of a large eDNA ocean survey (spanning 86 000 km² to depths of 500 m) to understand the abundance and distribution of Pacific hake (*Merluccius productus*), the target of the largest finfish fishery along the west coast of the USA. We sampled eDNA in parallel with a traditional acoustic-trawl survey to assess the value of eDNA surveys at a scale relevant to fisheries management. Despite local differences, the two methods yield comparable information about the broad-scale spatial distribution and abundance. Furthermore, we find depth and spatial patterns of eDNA closely correspond to acoustic-trawl estimates for hake. We demonstrate the power and efficacy of eDNA sampling for estimating abundance and distribution and move the analysis eDNA data beyond sample-to-sample comparisons to management relevant scales. We posit that eDNA methods are capable of providing general quantitative applications that will prove especially valuable in data- or resource-limited contexts.

## 1. Introduction

Environmental DNA, the DNA from target organisms collected from an environmental medium (e.g. soil or water), can reflect species in a wide range of terrestrial, aquatic, and marine habitats [1]. eDNA has the potential to revolutionize our understanding of natural communities by enabling rapid and accurate surveys of many species simultaneously [1]. At present, eDNA can efficiently survey species diversity and changes in community membership [2–4]. However, many natural resource questions depend upon quantitative estimates of abundance (e.g. fisheries or managing species of conservation concern), so eDNA must provide such information in order to be most useful [5]. While most studies find a positive relationship between eDNA concentrations and other survey methods (reviewed by Rourke *et al.* [6]), uncertainty about

the strength of the eDNA-abundance relationship due to the complexity of eDNA generation, transport, degradation and detection have limited the application of eDNA in many quantitative applications [5,7,8]. While the use of eDNA methods has grown exponentially from tens of publications in 2010 to many hundreds in 2020 [6,9], reflecting widespread adoption of eDNA technologies, basic questions about the characteristics of eDNA limit its practical application and slow its adoption in environmental management.

Rigorous, well-designed surveys underlie the successful management and conservation of wild populations. But field surveys are expensive—open-ocean surveys involve ship time costing tens of thousands of dollars per day—and are typically tailored to one or a few species. eDNA methods are appealing for open-ocean or other difficult-to-sample locations because sampling can be fast, standardized, non-lethal and detect many species simultaneously; sampling involves only the collection and processing of environmental samples [1]. Even modest improvements in sampling efficiency from current surveys can reduce the duration of surveys, yield substantial cost savings for focal species surveys, and free survey time to be reallocated to other understudied communities. However, such broad-scale implementation depends upon providing eDNA-based estimates of abundance at management-relevant scales [10–12]. To date, there have been no eDNA surveys conducted at sufficiently large scale to inform ocean fishery management, a field with many potential eDNA applications.

Observations of eDNA differ from observations derived from traditional methods (e.g. visual [13,14], capture [15,16] or acoustic [12] surveys) and the degree of agreement between individual samples of eDNA and traditional methods collected simultaneously often determines whether eDNA-based methods are viewed as successful or not [12,16]. However, eDNA observations arise from fundamentally different processes than observations from these traditional survey methods—most dramatically, by exponential amplification of DNA molecules in an environmental sample [17,18], but also because the distribution of eDNA itself in the environment is not identical to the distribution of its source organisms [5,7,8]. In the case of microbial eDNA, this distributional distinction is negligible, but for larger animals—such as fishes or marine mammals—it is not. Conceptually, fish are discrete, while the DNA traces they leave in the water are relatively continuous, blurring their environmental fingerprint over space and time [10]. For example, acoustic surveys of pelagic fishes reflect the patchy distribution of schooling fishes [19]. By comparison, we expect the associated eDNA to be distributed more evenly as a result of fish movement, the lag between shedding and decay processes, and water movement [7,8,11]. Thus, simple sample-level comparisons between eDNA and other survey methodologies are a poor method for determining the usefulness of eDNA surveys. Understanding the ecology of eDNA [7] makes possible an honest assessment of the potential uses and limitations of eDNA for applied environmental problems, and allows each data stream to be used to its best advantage.

Here, we leverage a spatially extensive eDNA survey of the oceans—spanning over 86 000 km$^2$ and to depths of 500 m—to document the empirical patterns of eDNA for a commercially important and abundant fish species, Pacific hake (*Merluccius productus*). Hake is a semi-pelagic schooling species and is among the most abundant fish species in the California Current Ecosystem [20,21]. They support a large and important fishery along the Pacific coasts of USA and Canada with coastwide catches in excess of 400 000 t annually and ex-vessel value in excess of $60 million in recent years [21]. Hake are a key component of the California Current ecosystem as both predator and prey, migrating to the surface at night and back to mid-water depths during the day [22]. Seasonally, adults migrate between southern spawning areas and northern foraging areas [20,22,23]. The rich datasets available for hake provide an opportunity to rigorously compare available information from traditional surveys with eDNA using parallel statistical models that relate observations from each data type to quantitative indices of abundance.

We investigate large-scale and depth-specific spatial patterns of hake DNA in the open ocean using a quantitative PCR assay targeting the 12S mitochondrial gene region [24]. We show how eDNA can be aggregated to provide a depth-integrated index of hake abundance comparable to acoustic-trawl survey results used for fisheries stock assessments [21]. The spatial–statistical model we use is a first for eDNA in the ocean and makes results for eDNA surveys comparable to other methods used in quantitative natural-resources management [25,26]. We derive metrics of the species' spatial and depth distribution and investigate the relative precision of the eDNA and acoustic-trawl surveys. Our results show that eDNA analyses can provide important information about species abundance and distribution at management-relevant scales, provide relatively straightforward opportunities for supplementing existing surveys, and open the door for providing quantitative information for additional species that are currently un- or under-studied.

## (a) Material and methods

### (i) Field sampling and processing for eDNA

We collected eDNA samples during the 2019 U.S.–Canada Integrated Ecosystem & Acoustic-Trawl Survey for Pacific hake aboard the NOAA Ship *Bell M. Shimada* from 2 July to 19 August [27] including waters from 38.3°N to 48.6°N along the Pacific coast of the USA (123°W to 126.5°W longitude). Detailed collection protocols and all laboratory analyses including information on sample preservation and extraction, primers (12S primer description, specificity and sensitivity testing, and other aspects), qPCR protocols, voucher specimens and all other steps are provided in Ramón-Laca *et al.* [24]. We briefly summarize those protocols here.

We collected seawater from up to six depths (3, 50, 100, 150, 300 and 500 m) at 186 stations where a conductivity, temperature and depth (CTD) rosette was deployed. These stations were spread across 36 acoustic transects (figure 2). We included 1769 individual water samples collected at 892 depth-station combinations (a small number of samples were contaminated or lost during processing). In total, 710 depth-stations were collected at 50 m deep or deeper. Two replicates of 2.5 l of seawater were collected at each depth and station from independent Niskin bottles attached to a CTD rosette. Water samples from 3 m were collected from the ship's saltwater intake line but processed identically to Niskin samples. Nearly all CTD casts, and therefore water collection, for eDNA occurred at night while acoustic-trawl sampling (see below) took place during daylight hours.

To account for possible contamination, negative sampling controls were collected routinely by filtering 2 l of distilled water from either the onboard evaporator or from distilled water brought from the laboratory for this purpose (N = 49 in total). Both

Niskin collected and control samples were filtered immediately using 47 mm diameter mixed cellulose ester sterile filters with a 1 µm pore size using a vacuum pump. The filters were stored at room temperature in Longmire's buffer until DNA extraction [28]. We detected low levels of hake contamination in control samples with most negative controls having average estimated DNA concentrations of 44 copies l$^{-1}$ which is slightly larger than the detection threshold of 20 copies l$^{-1}$ (see electronic supplementary material, figures S3 and S4), but below most estimated hake DNA concentrations from field samples.

The DNA was extracted using a modified phenol : chloroform method with a phase lock to increase the throughput and yield [24]. Quantification of Pacific hake was performed by qPCR using a specific TaqMan assay on a QuanStudio 6 (Applied Biosystems) that included an internal positive control (IPC) of the reaction to account for PCR inhibition. Any delay of more than 0.5 cycles from the IPC at the non-template controls of the PCR was considered inhibition. For inhibited samples, we used a 1 : 5 dilution in subsequent analyses. A subset of samples had a final wash with an incorrect concentration ethanol (30% ethanol instead of 70% ethanol). These samples had reduced hake DNA concentrations and we accounted for samples with the improper wash in our statistical model below (see electronic supplementary material for more details).

### (ii) Spatial eDNA model

We developed a Bayesian state-space framework for modelling DNA concentration in the coastal ocean. State-space models separate the true biological process from the methods used to observe the biological process [29,30]. In our case, the biological process of interest is the spatial- and depth-specific pattern of hake eDNA. Let $D_{xyd}$ be the true, but unobserved, concentration of hake DNA (DNA copies l$^{-1}$) present at spatial coordinates $\{x, y\}$ (northings and eastings, respectively, in kilometres) and sample depth $d$ (metres). We model the DNA concentration as a spatially smooth process at each depth sampled ($d = 3$, 50, 100, 150, 300 or 500 m) and linear on the $\log_{10}$ scale,

$$\log_{10} D_{xyd} = \gamma_d + s(b) + t_d(x, y), \tag{1.1}$$

where $\gamma_d$ is the spatial intercept for each depth, $s(b)$ indicates a smoothing spline as a function of bottom depth in metres ($b$), and $t_d(x, y)$ is a tensor-product smooth that provides an independent spatial smooth for each depth. We use cubic regression splines for both univariate and tensor-product smoothes. We investigated a range of knot densities for smoothes in preliminary investigations (see electronic supplementary material).

From the process model in equation (1.1), we construct a multi-level observation model. First, we model the DNA concentration in each Niskin bottle $i$, as a random deviation from the true DNA concentration at that depth and location and include three offsets to account for variation in the processing of eDNA extracted from Niskin bottles.

$$\log_{10} E_i = \log_{10} D_{xyd} + \delta_i + \log_{10} V_i + \log_{10} I_i + \mathbf{I}\omega \tag{1.2}$$

and

$$\delta_i \sim \text{Normal}(0, \tau_d), \tag{1.3}$$

where $V_i$ is the proportion of 2.5 l filtered from Niskin $i$ (in nearly all cases $V_i = 1$), $I_i$ is the known dilution used on sample $i$ to eliminate PCR inhibition, and $\mathbf{I}\omega$ is an estimated offset for an ethanol wash error. Here, $\mathbf{I}$ is an indicator variable where $\mathbf{I} = 1$ for affected samples and $\mathbf{I} = 0$ otherwise (see also electronic supplementary material).

When using qPCR, we do not directly observe eDNA concentration, we observe the PCR cycle at which each sample can be detected (or if it was never detected). We use a hurdle model to account for the fact that there is a probabilistic detection threshold (the PCR cycle of amplification is detected $G = 1$ or is not observed

$G = 0$). Conditional on being detected, we observe the PCR cycle ($C$) as a continuous variable that follows a $t$ distribution,

$$G_{ijr} \sim \text{Bernoulli}(\phi_{0j} + \phi_{1j} \log_{10} E_i) \tag{1.4}$$

and

$$C_{ijr} \sim T(\nu, \beta_{0j} + \beta_{1j} \log_{10} E_i, \eta) \quad \text{if } G_{ijr} = 1, \tag{1.5}$$

Here $j$ indexes the PCR plate on which sample $i$ and replicate $r$ were run. We conducted three PCR reactions for each $E_i$. We fix the degrees of freedom for the $t$-distribution ($\nu = 3$) to allow for heavy-tailed observations and the parameter $\eta$ is a scale parameter that controls the dispersion of the distribution. Note that there are different intercept ($\phi_{0j}$, $\beta_{0j}$) and slope ($\phi_{1j}$, $\beta_{1j}$) parameters for each PCR plate to allow for among-plate variation in amplification. See the electronic supplementary material for additional components of the statistical model. We use diffuse prior distributions for all parameters (electronic supplementary material, table S1).

### (iii) Acoustic-trawl data

In parallel with water collection for eDNA, we incorporated data on hake biomass derived from the contemporaneously collected data [27], consisting of 57 acoustic transects totalling 4483 km in length. In total, 45 midwater trawls were deployed that provide information on the age, size and therefore signal strength of hake [21,27]. Methods for converting raw acoustic and trawl data to biomass concentrations can be found in [20,21] and references therein. We used derived estimates of biomass concentration (t km$^{-2}$) for hake ages 2 and older that integrate the biomass in the water column between depths of 50 and 500 m in all analyses. All acoustic data and associated trawls were collected during daylight hours. Therefore there was a lag between collection of acoustic-trawl and eDNA data, though for nearly all cases these were separated by less than 24 h. The temporal separation of eDNA and acoustic-trawl sampling precluded direct comparisons at the single-sample level.

### (iv) Spatial acoustic-trawl model

In parallel with the model for qPCR data, we estimated a spatial model for the hake biomass derived from the acoustic-trawl survey. The biomass index created from the acoustic-trawl data for the entire survey area (34.4° N to 54.7° N) is used in stock assessments that determine the allowable catch and allocation of hake catch for fleets from the USA, Canada and Tribal Nations [21]. As the eDNA samples only cover a portion of this range (38.3° N to 48.6 N), we used the biomass observations within this latitudinal range to generate spatially smooth estimates of biomass. Acoustic transects are divided into 0.926 km (0.5 nm) segments and the biomass (age 2 and older) concentration within each segment is used as data [21].

Unlike the eDNA data, age-specific biomass estimates are available only as a biomass integrated across the entire water column (from depths of 50 to 500 m). We fit a Bayesian hurdle model using a form similar to the eDNA, modelling biomass concentration ($F_{xy}$; units: t km$^{-2}$) using two separate spatial submodels: (a) the probability of occurrence and (b) abundance conditional on the presence of hake. We model both components as a function of bottom depth (smooth) and a spatial smooth,

$$H_{xy} \sim \text{Bernoulli}(\text{logit}^{-1}(\zeta_H + s_H(b) + t_H(x, y)) \tag{1.6}$$

and

$$F_{xy} \sim \text{LogNormal}(\zeta_F + s_F(b) + t_F(x, y) - 0.5\kappa^2, \kappa)$$
$$\text{if } H_{xy} = 1, \tag{1.7}$$

where $H_{xy}$ is 1 if the observed biomass concentration is non-zero and zero otherwise. In this formulation, $\zeta$ is the spatial intercept for each

*Proc. R. Soc. B* **289**: 20212613

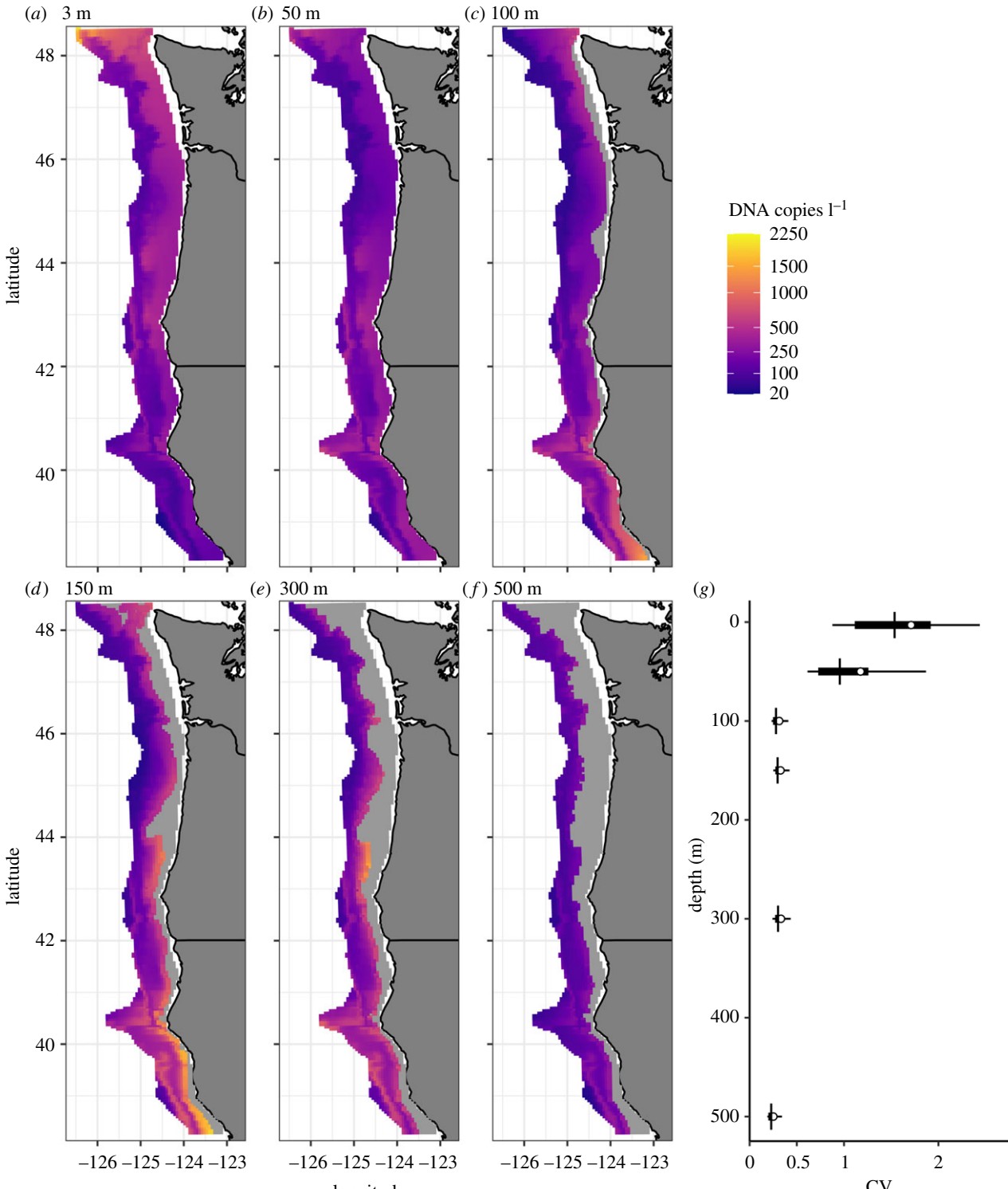

**Figure 1.** Predicted DNA concentration for six water depths shows clear spatial patterning in DNA concentration ((*a*–*f*); posterior mean). (*g*) Uncertainty around the posterior mean for each water depth as measure by the coefficient of variation. The distribution of CV among all projected 25 km$^2$ grid cells are shown (mean (circle), median (vertical line), 50% and 90% CI shown). (Online version in colour.)

model component, $s(b)$ indicates a smoothing spline of as a function of bottom depth in metres ($b$), and $t(x, y)$ is a tensor-product smooth over latitude and longitude. $\kappa$ is the standard deviation of the positive observations on the log scale. Electronic supplementary material, table S2 provides the prior distributions for this model.

### (v) Model estimation

We implemented both the eDNA and acoustic-trawl models using the Stan language as implemented in R (*Rstan*). All relevant code and data are provided in the electronic supplementary material

and data repository [31]. For the eDNA model, we ran four MCMC chains using 1500 warm up and 9000 sampling iterations. For the acoustic-trawl model, we ran four MCMC chains using 1200 warm up and 3000 sampling iterations.

We used traceplots and $\hat{R}$ diagnostics to confirm convergence ($\hat{R} < 1.01$ for all parameters)—there were no divergent transitions in the sampling iterations. To generate design matrices necessary for estimating covariate effects, we used the R package *brms* [32,33]. We use diffuse prior distributions for all parameters (electronic supplementary material, table S1). Posterior summaries of parameters can be found in the electronic supplementary material.

Proc. R. Soc. B 289: 20212613

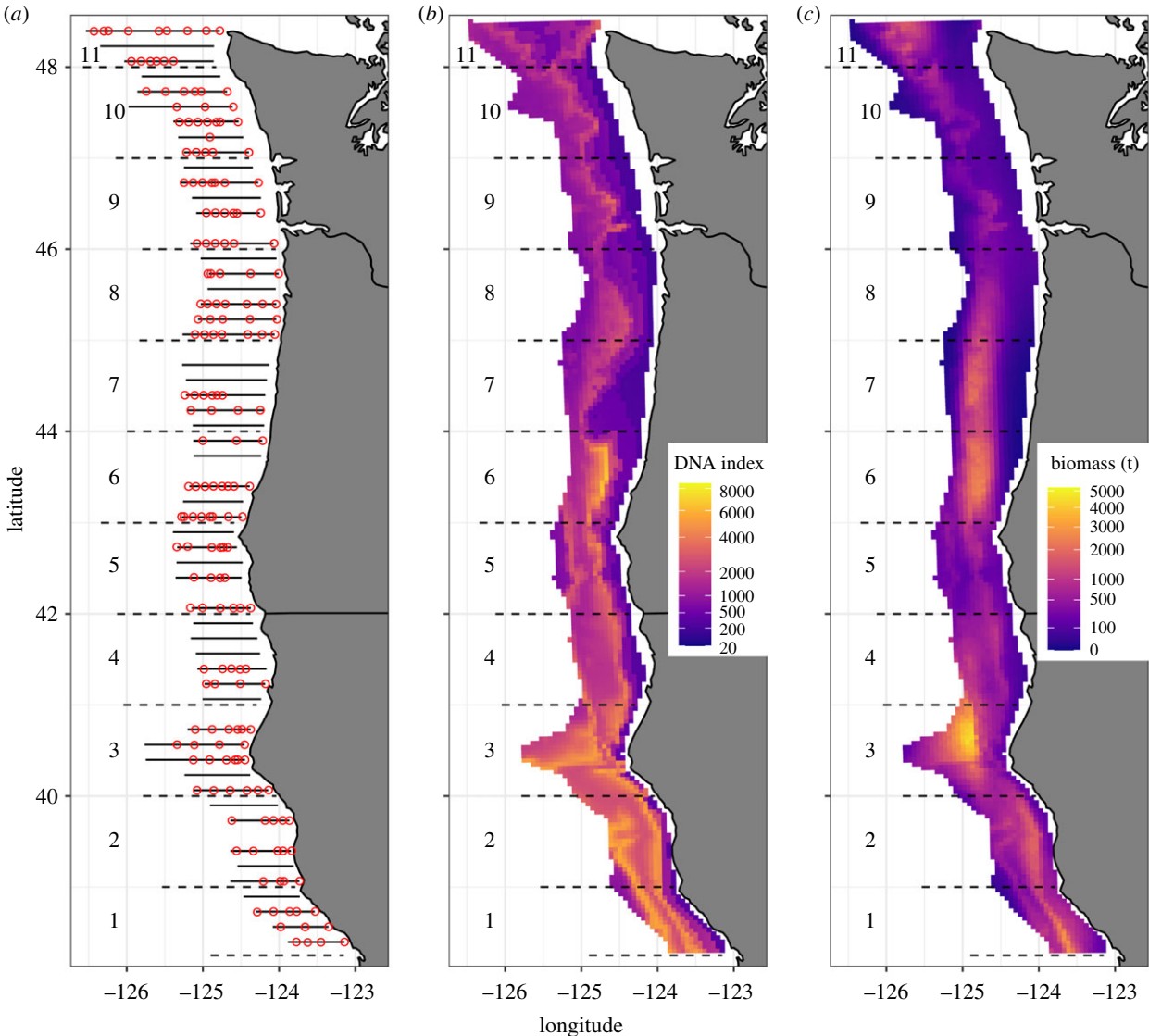

**Figure 2.** Survey locations for 2019 ((*a*) circles show eDNA sampling locations, lines show acoustic transects), depth-integrated index of hake DNA (*b*) and hake biomass from acoustic surveys (*c*). Both DNA and acoustic estimates are mean predicted values projected to a 5 km grid and include information between 50 and 500 m deep. All panels show one degree latitudinal bins (numbered; separated by dashed lines) used to aggregate abundance estimates over larger spatial scales (figure 3). (Online version in colour.)

### (vi) Coordinate systems, covariates and spatial predictions

We generated 5 km resolution gridded maps for both the acoustic-trawl and eDNA models to enable direct comparisons between models. This vector-based grid was developed and used by others [34] for interpolating various spatial models and uses a custom coordinate reference system that conserves area and distance reasonably well across the west coast of the USA (electronic supplementary material) and was a suitable resolution for the purposes of our analyses.

To create spatial predictions for both eDNA and acoustic-trawl models, we took 4000 draws from the joint posterior and generated predictions for the centroid of each grid cell. We calculated posterior means and uncertainty bounds among posterior draws. For the eDNA model we made projections for $D_{xyb}$; we do not present results from including additional observation processes on top of the estimated DNA concentrations in the main text. We generated posterior predictive distributions for other model diagnostics checks (see electronic supplementary material).

### (vii) Creating an eDNA index

Our model provides direct predictions for hake DNA concentration at depths of 50, 100, 150, 300 and 500 m. The model lacks a term to directly make predictions to water depths other than those that were observed. Therefore, to produce an index spanning depths of 50–500 m, we equally weighted depths between 50 and 500 m using linear interpolation between the closest depths. We used posterior predictions at each depth to provide predicted DNA densities at 200, 250, 350, 400 and 450 m for each 5 km grid cell. Because some spatial locations have depths of less than 500 m, we only include predicted DNA concentrations to a depth appropriate for the bathymetry (e.g. a location with a depth of 180 m only includes values from 50, 100 and 150 m). We sum across all depths (between 50 and up to 500 m) to generate a depth-integrated index of hake DNA. This index will be proportional to the hake DNA found in the water column. However, as we are only summing across discrete depths, not integrating values across the entire water column nor multiplying by the total water volume within each grid cell, the absolute value of the index will depend upon the number of discrete depths we use. We refer to this as an eDNA abundance index to differentiate it from predictions for specific depths.

We compare estimates from the acoustic-trawl with the eDNA index using Pearson product-moment correlations. We compare predictions from the methods at the scale of 25 km² grid cells and after aggregating estimates from each method into 1° latitudinal bins (for a total of 11 bins; figure 2).

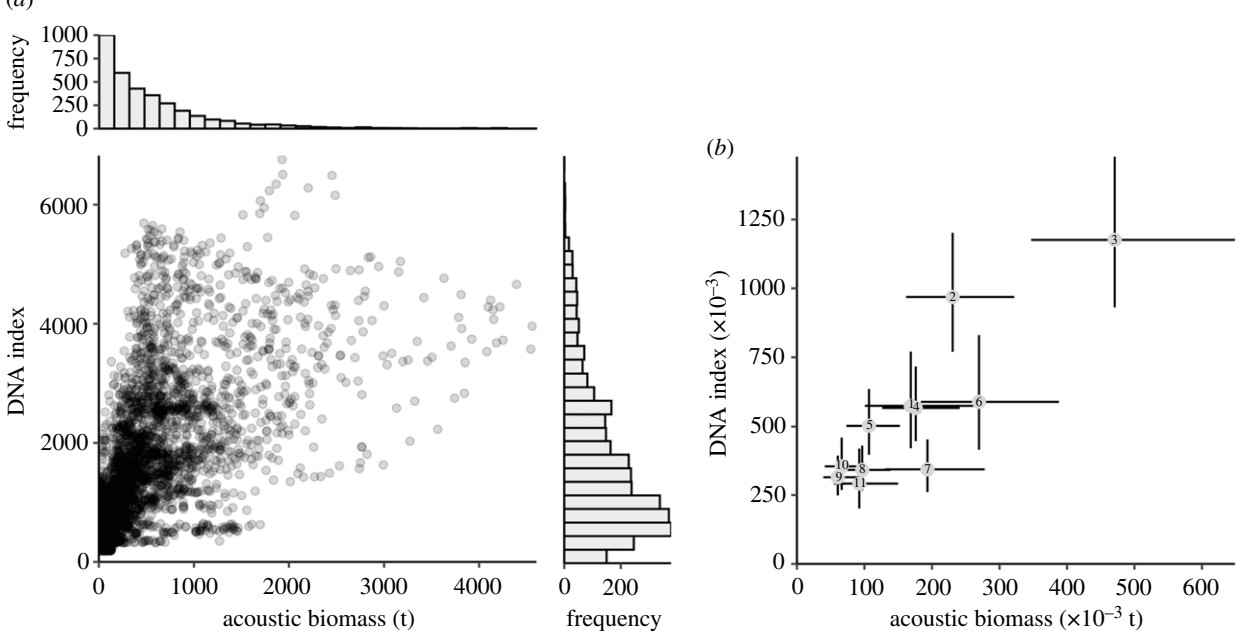

**Figure 3.** Pairwise comparison between DNA and acoustics-derived biomass. (*a*) Posterior mean prediction from each method among the 3455 grid cells of 25 km² and includes the marginal histogram of posterior mean values for each method (correlation of posterior mean [90% CI]; $\rho = 0.55[0.53, 0.57]$). (*b*) Correlation between methods among the 11, one degree latitude bins (posterior mean [90% CI] shown; $\rho = 0.88[0.65, 0.96]$). Numbers indicate regions identified in figure 2.

## (b) Results

Hake were detected throughout the survey region (figures 1 and 2), but hake DNA was far more commonly detected than the acoustic-trawl signature of hake. As expected for a patchily distributed species, acoustic-trawl sampling identified hake biomass in a minority of 0.983 km long transect segments (1764 of 4841; 36%). By contrast, hake eDNA was detected in 94% of water samples (non-zero concentrations of hake DNA were quantified in 1670 of 1769 samples of 2.5 l) and 98% of sampling stations (875 of 892 stations), reflecting a considerable increase in detection of eDNA signal relative to the acoustic-trawl detections.

Hake DNA in the study area varied substantially (estimated DNA concentration of individual samples ranged from below detection (<20 copies l⁻¹) to greater than 40 000 copies l⁻¹). DNA concentrations at stations—there are two water samples at each depth-station—varied strongly with depth, with high estimated DNA concentrations at 150 m (grand mean[range] = 377[36 − 1701] copies l⁻¹) and 300 m depth (306[69 − 1567]). DNA concentration at stations declined at both shallower (e.g. 50 m: 180[44 − 535] copies l⁻¹) and deeper depths (500 m: 144[46 − 382] copies l⁻¹). Hake DNA showed notable spatial patterns, peaking along the continental shelf break and south of the Oregon–California border at 42°N (figure 1).

There were also striking patterns in the variation in Hake DNA concentration with depth (figure 1). Specifically, deep stations (100, 150, 300 and 500 m) had relatively low uncertainty with regard to hake concentration (median coefficient of variation (CV) of approx. 0.3), whereas the median CV for both 3 and 50 m depths was larger than 1. Large CVs indicate both large bottle to bottle variation in DNA concentration within a station and substantial variation in hake eDNA concentration among nearby sampling locations (figure 1 and electronic supplementary material, figure S8).

We combined DNA information between 50 and 500 m to produce both a spatially smooth, depth-integrated estimate of hake DNA concentration (figure 2b). Separately, we generated a spatially smooth estimate of age 2+ biomass from the acoustic-trawl survey (figure 2c). The eDNA abundance index showed strong spatial patterning with highest values along the continental shelf break with notable peaks in central California and Oregon waters. In contrast,

acoustic-trawl observations were highly spatially variable—a common feature observed in acoustic surveys [35]—with high hake density and others with very low density in close proximity (see also electronic supplementary material, figure S17). At the scale of individual 25 km² grid cells, eDNA and acoustic-trawl surveys were modestly correlated ($\rho = 0.55[0.53, 0.57]$, Pearson product-moment correlation on posterior mean prediction [90% CI]; figure 3) but there is considerable scatter in the relationship. Large eDNA values never occurred at locations that had very low acoustic-trawl biomass, but very high acoustic-trawl estimates corresponded to moderate values of eDNA. Notably, acoustic-trawl biomass estimates had a very right-skewed distribution across the 3455 ocean cells of 25 km² considered—most values were near zero with very few high values—while eDNA values were decidedly less skewed (figure 3). Taken together, these observations again suggest a smoother distribution of eDNA information relative to the patchier acoustic-trawl detections.

When aggregated to 1° latitude bins, the correlation between eDNA and acoustic-trawl increased substantially ($\rho = 0.88[0.65, 0.96]$; figure 3) with acoustic-trawl and eDNA scaling approximately linearly. Such increased correlation is not dependant upon the spatial groupings in figure 3 (see electronic supplementary material, figures S15 and S16 for results from an alternate spatial grouping). At this scale, eDNA and acoustic-trawl provide nearly equivalent information about relative biomass. At a coast-wide scale, the uncertainties (CVs) of the acoustic-trawl estimate and eDNA index were nearly identical (both 0.09). This similarity occurred despite the eDNA only being collected at 186 locations, whereas the acoustic-trawl data includes 4841 acoustic transect segments and 45 midwater trawls to determine age- and length-structure of the hake.

Finally, the two methods produced nearly identical latitudinal distributional estimates as measured by centre of gravity (median value within the projection range) and cumulative distribution (90% CIs overlapping for the entire latitudinal range; figure 4). Furthermore, averaged across space, hake DNA concentrations were highest along the continental shelf break (bottom depths between 125 and 400 m) and at water depths between 150 m and 300 m (figure 4c). All of these observations are consistent with published descriptions of hake depth and habitat preferences [20,21,36].

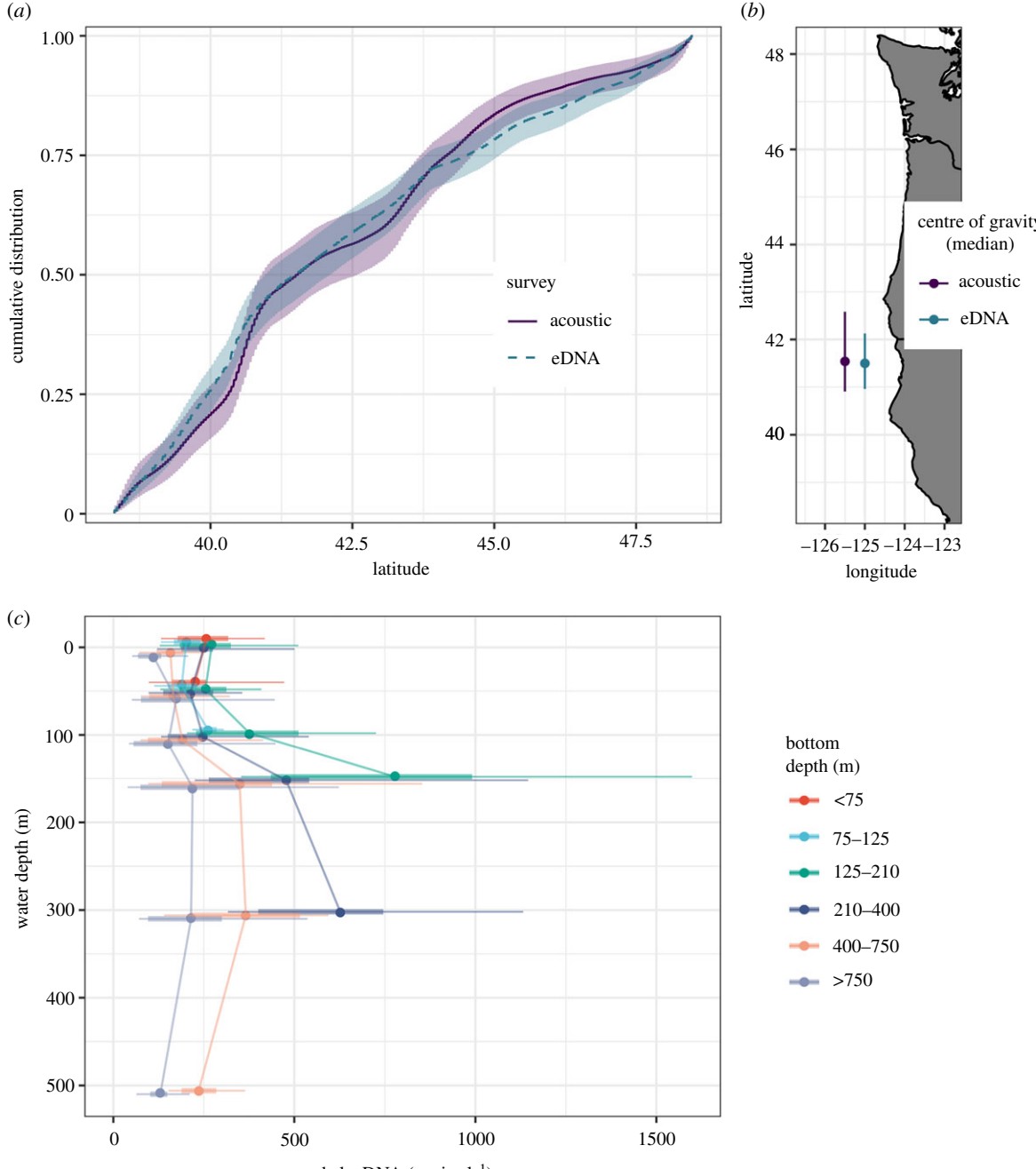

**Figure 4.** Estimates of distribution of Pacific hake. (*a*) Cumulative distribution between 38.3 and 48.6°N (posterior means, 90% CI). (*b*) Centre of gravity (median of distribution) for each method (posterior means and 90% CI; only areas within the projection grid are included in this calculation; see figures 1 and 2). (*c*) Posterior estimates of hake DNA concentration at each station-depth combination by the water depth sampled and categories of the depth of the bottom. The distribution of mean DNA concentration among station-depths (mean, interquartile range and 90% CI among station-depths). Bottles at a sample location become increasingly similar at deeper sampling depths. (Online version in colour.)

## (c) Discussion

Ocean surveys are often used to generate large-scale, quantitative indices of species' abundances. At the spatial scale relevant to management for hake along the US west coast—our survey region encompasses the majority of habitat for the Pacific hake stock—analysis of water samples taken for eDNA provides comparable indices of hake biomass to acoustic-trawl surveys despite far fewer eDNA observations. While other efforts have developed quantitative methods for eDNA within rivers [10], lakes [15,37], estuaries [11] and nearshore marine habitats [12], we produce a large-scale study that can serve as a template for using eDNA to determine abundance and species distributions with clear practical applications to both conservation and fisheries. Importantly, our analysis demonstrated the value in analyses that push beyond simple sample-to-sample comparisons between eDNA and other alternate sampling methods

to make inferences at the population-scale. The spatial scale investigated here (on the order of tens of thousands of square kilometres) is roughly comparable to the scale at which most large ocean fisheries are managed both in the USA and internationally, suggesting eDNA approaches can begin to be broadly adopted for that purpose.

The kind of spatial–statistical model we report here brings eDNA analysis in line with the methods currently used in quantitative natural-resources management [e.g.27,38]. Despite the clear differences in biological processes producing eDNA signals versus acoustic trawl signals, these distinct datasets are both subject to rigorous analytical methods. We emphasize that eDNA data here are processed independently from acoustic-trawl data; no information from the acoustic-trawl informs eDNA or vice versa. Thus, the implementation of eDNA surveys provides a second survey of abundance

for hake without requiring any additional days at sea, and should provide improved precision for estimated fish abundance when the two indices are incorporated into a stock assessment. Additionally, the eDNA samples are archived and can be used to investigate other species in future analyses. eDNA holds unprecedented potential for improving the precision of abundance surveys, particularly when conducted in concert with existing surveys.

For determining an index of abundance over a very large area, we assert that eDNA works well because the concerns about the impact of DNA transport, degradation and other processes [5,7,8] are negligible. Hake DNA present within our survey boundaries was generated by hake present within the survey area; oceanographic processes like currents or upwelling are not of sufficient magnitude to transport meaningful amounts of water into or out of the survey domain on the timescale at which eDNA degrades [8]. Similarly, rates of DNA degradation are expected to be consistent across our sampling domain—cool, off-shore, oceanic waters below 50 m with relatively little among-sample variation in temperature, salinity, and other covariates identified as important for degradation [39,40]. Such population closure and constant rate assumptions are reasonable [see also 12] and allow us to treat eDNA observations as analogous to other traditional sampling methods. We note that our modelling framework provides the flexibility to directly include relevant covariates into the observation model to account for relevant DNA processes if and when such information becomes available (see Material and methods and electronic supplementary material). For hake, our eDNA results match available geospatial (figures 2 and 4) and depth-specific patterns of hake abundance [20,25] (figure 4) from other methods, strongly suggesting our assumptions are reasonable and justified. eDNA approaches may be less effective in applications focused on smaller temporal and spatial scales such as detailed habitat-association studies where the precise locations of individuals are required (but see [41,42]).

Many challenges to implementing eDNA surveys remain. Surveys are primarily valuable because they inform temporal trends; most surveys, particularly those of marine species, are not used as measures of absolute abundance but as indices of abundance relative to previous years [21,38]. It will accordingly require years to accumulate the kinds of eDNA-based time series that parallel those used in current management and can be used in a stock assessment context. Furthermore, there are additional data streams needed for management applications that are not currently possible from eDNA. For example, physical specimens are needed to document age, size, sex and condition, all of which cannot be extracted from eDNA at present, though these are active areas of research [43,44]. At present, eDNA approaches should be regarded as supplementing existing surveys, not replacing them.

Despite these limitations, the characteristics of eDNA surveys have several advantages. First, the samples collected and analysed here for hake can be re-analysed for other species. Analyses using species-specific qPCR should provide similar quantitative data for additional species. DNA metabarcoding approaches can detect many species simultaneously [1], but metabarcoding results are difficult to link to abundance or biomass [17,18]. Second, surveys of eDNA provide the potential for large-scale replication and high precision because they only involve collecting water; as many replicate

samples as desired can be collected, enabling researchers to target and achieve a desired level of precision. Such replication is often not possible for other sampling methods that involve capturing individuals. For example, repeatedly trawling a particular location will deplete the fish present, and therefore such repeated sampling is generally not helpful for estimating abundance. In theory, there are few limits on replication using eDNA and our results indicate that the amount of small-scale variation between water samples declines with depth (figure 1g and electronic supplementary material, figure S9), suggesting that the amount of statistical noise and therefore the amount of sampling needed may vary concomitantly. It is wholly unknown if other marine species will exhibit similar depth-specific patterns of variability to those observed in hake, though we hypothesize that the patterns observed may be related to the diel vertical migration patterns of hake.

We developed and applied our eDNA approach to Pacific hake because of its broad geographical range, economic importance and decades of associated survey information. The ability of eDNA to provide indices of abundance and distribution lend strong support for the applicability of eDNA methods to the unstudied majority of species in ocean ecosystems. We believe eDNA will be particularly valuable for understanding future changes in distribution of hake as well as other species, and future work will connect eDNA surveys and oceanographic variables to understand shifts in species distributions.

Data accessibility. Additional methodology and analysis are provided in the electronic supplementary material [45] and data and code are available from the Dryad Digital Repository: https://doi.org/10.5061/dryad.n2z34tmzf [31].

Authors' contributions. A.O.S.: conceptualization, formal analysis, funding acquisition, methodology, software, supervision, visualization, writing—original draft, writing-review and editing; A.R.-L.: data curation, investigation, methodology; A.W.: investigation, methodology, resources; J.C.: methodology, project administration, supervision, writing—review and editing; D.C.: data curation, writing-review and editing; B.E.F.: data curation, visualization, writing—review and editing; R.P.K.: visualization, writing-original draft, writing—review and editing; S.L.P.S.: conceptualization, funding acquisition, project administration, writing—review and editing; R.T.: data curation, methodology, writing—review and editing; K.M.N.: conceptualization, funding acquisition, project administration, supervision, writing—review and editing; L.P.: conceptualization, project administration, supervision, writing—review and editing.

All authors gave final approval for publication and agreed to be held accountable for the work performed therein.

Competing interests. We declare we have no competing interests.

Funding. Funding for this project was provided by the National Marine Fisheries Service's Genomic Strategic Initiative.

Acknowledgements. Special thanks to S. Allen, B. Dewees, J. Witmer and J. Davis for assistance during sample collection and the captain and crew of the NOAA ship Bell M. Shimada for overall support during the survey cruise. A. Billings, J. Pohl and members of the Fisheries Engineering and Acoustic Technologies team provided logistical and analytical support for both the acoustic and eDNA components of the survey. E. Iwamoto and members of the molecular genetics lab at the Northwest Fisheries Science Center supported laboratory work. A. Berger, M.J. Ford, J.F. Samhouri and two anonymous reviewers provided helpful comments on earlier versions of the manuscript.

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
