## [Peer Review File · Proceedings of the Royal Society B: Biological Sciences]

Review History

RSPB-2021-2508.R0 (Original submission)

Review form: Reviewer 1

Recommendation

Accept with minor revision (please list in comments)

Scientific importance: Is the manuscript an original and important contribution to its field?

Excellent

General interest: Is the paper of sufficient general interest?

Excellent

Quality of the paper: Is the overall quality of the paper suitable?

Good

Is the length of the paper justified?

Yes

Should the paper be seen by a specialist statistical reviewer?

Yes

Do you have any concerns about statistical analyses in this paper? If so, please specify them explicitly in your report.

No

It is a condition of publication that authors make their supporting data, code and materials available - either as supplementary material or hosted in an external repository. Please rate, if applicable, the supporting data on the following criteria.

Is it accessible?

Yes

Is it clear?

No

Is it adequate?

Yes

Do you have any ethical concerns with this paper?

No

Comments to the Author

Please see the attached file for manuscript comments. I am happy to expand or answer any questions related to my comments. Good luck!

Review form: Reviewer 2

Recommendation

Excellent

Scientific importance: Is the manuscript an original and important contribution to its field?

Good

General interest: Is the paper of sufficient general interest?

Good

Quality of the paper: Is the overall quality of the paper suitable?

Good

Is the length of the paper justified?

Yes

Should the paper be seen by a specialist statistical reviewer?

No

Do you have any concerns about statistical analyses in this paper? If so, please specify them explicitly in your report.

No

It is a condition of publication that authors make their supporting data, code and materials available - either as supplementary material or hosted in an external repository. Please rate, if applicable, the supporting data on the following criteria.

Is it accessible?

Yes

Is it clear?

Yes

Is it adequate?

Yes

Do you have any ethical concerns with this paper?

No

Comments to the Author

The paper is extremely relevant for conservation. It stems from the fact that eDNA has become an efficient method to assess species diversity and changes in community with the potential to greatly improve our understanding of natural communities while it remains unclear whether eDNA signals can provide quantitative metrics of abundance to support management. The study is based on the results of a large ocean survey (spanning 86,000 km² to depths of 500m) and is focused on the abundance and distribution of Pacific hake (*Merluccius productus*) along the west coast of the United States. The knowledge available for hake provides an opportunity to rigorously compare available information from traditional surveys with eDNA assessment. The paper is well written and suitable for the journal and it could be accepted on its present form. My only questions and suggested revisions are the following:

- among the most significant results there is the assessment of hake DNA variability in the study area which varied substantially with depth, with the highest concentrations between 100m and 300m depth (which I believe is consistent with the species preferred habitat) and concentrations lower and more homogeneous at depth than near the surface. I was wondering whether the fact that the most of water collection for eDNA occurred at night may have also an influence in this respect. Perhaps the authors may want to add this in their discussion;
- about the eDNA index that was created for the purpose of the spatial analysis, the authors explain that they have generated a depth-integrated index of hake DNA summing the values across all depths and not integrating values across the entire water column or multiplying by the total water volume within each grid cell so that the absolute value of the index depends upon the number of discrete depths at each location. I have two questions in this respect (same questions that I would expect also the readers may have): why the authors decided to use this index instead of the posterior predictions at each depth provided at 200, 250, 350, 400, and 450m for each 5km grid cell, and secondly, given that some locations spatial locations had depths lower than 500m, why they did not standardise the index to a depth common to all the locations?

Decision letter (RSPB-2021-2613.R0)

21-Jan-2022

Dear Dr Shelton:

Your manuscript has now been peer reviewed and the reviews have been assessed by an Associate Editor. The reviewers' comments (not including confidential comments to the Editor) and the comments from the Associate Editor are included at the end of this email for your reference. As you will see, the reviewers and the Editors have raised some concerns with your manuscript and we would like to invite you to revise your manuscript to address them.

We do not allow multiple rounds of revision so we urge you to make every effort to fully address all of the comments at this stage. If deemed necessary by the Associate Editor, your manuscript will be sent back to one or more of the original reviewers for assessment. If the original reviewers

are not available we may invite new reviewers. Please note that we cannot guarantee eventual acceptance of your manuscript at this stage.

Research ethics:

Use of animals and field studies:

It is a condition of publication that you make available the data and research materials supporting the results in the article. Please see our Data Sharing Policies (<https://royalsociety.org/journals/authors/author-guidelines/#data>). Datasets should be deposited in an appropriate publicly available repository and details of the associated accession number, link or DOI to the datasets must be included in the Data Accessibility section of the article (<https://royalsociety.org/journals/ethics-policies/data-sharing-mining/>). Reference(s) to datasets should also be included in the reference list of the article with DOIs (where available).

Please submit a copy of your revised paper within three weeks. If we do not hear from you within this time your manuscript will be rejected. If you are unable to meet this deadline please let us know as soon as possible, as we may be able to grant a short extension.

Best wishes,

Professor Gary Carvalho

Associate Editor

Comments to Author:

Your manuscript has now been evaluated by two expert reviewers. As you will see, both reviewers were positive about the manuscript and noted its value to address the important question of how well environmental DNA measurements, in this case of fish, correlate to traditional measurements. I can corroborate these reviews and affirm that the manuscript would in principle be appropriate for publication in Proceedings B. The reviewers did, however, highlight multiple sections of the manuscript that could be improved for clarity and more detail. I concur that incorporating some or most of these suggestions would improve the manuscript and better allow aspects of the work to be reproduced. Specifically, the first reviewer provided a lengthy review, suggesting improvements to the title, abstract, introduction, and methods, which should improve readability and reproducibility. The second reviewer provided suggestions or questions about the results and discussion, which could be clarified in a revision. Please make sure to respond to each reviewer comment in a point-by-point response and consider including a tracked changes version along with a clean version of your manuscript with your resubmission.

Reviewer(s)' Comments to Author:

Referee: 1

Comments to the Author(s)

Please see the attached file for manuscript comments. I am happy to expand or answer any questions related to my comments. Good luck!

Referee: 2

Comments to the Author(s)

The paper is extremely relevant for conservation. It stems from the fact that eDNA has become an efficient method to assess species diversity and changes in community with the potential to greatly improve our understanding of natural communities while it remains unclear whether eDNA signals can provide quantitative metrics of abundance to support management. The study is based on the results of a large ocean survey (spanning 86,000 km² to depths of 500m) and is focused on the abundance and distribution of Pacific hake (*Merluccius productus*) along the west coast of the United States. The knowledge available for hake provides an opportunity to

rigorously compare available information from traditional surveys with eDNA assessment. The paper is well written and suitable for the journal and it could be accepted on its present form. My only questions and suggested revisions are the following:

- among the most significant results there is the assessment of hake DNA variability in the study area which varied substantially with depth, with the highest concentrations between 100m and 300m depth (which I believe is consistent with the species preferred habitat) and concentrations lower and more homogeneous at depth than near the surface. I was wondering whether the fact that the most of water collection for eDNA occurred at night may have also an influence in this respect. Perhaps the authors may want to add this in their discussion;

- about the eDNA index that was created for the purpose of the spatial analysis, the authors explain that they have generated a depth-integrated index of hake DNA summing the values across all depths and not integrating values across the entire water column or multiplying by the total water volume within each grid cell so that the absolute value of the index depends upon the number of discrete depths at each location. I have two questions in this respect (same questions that I would expect also the readers may have): why the authors decided to use this index instead of the posterior predictions at each depth provided at 200, 250, 350, 400, and 450m for each 5km grid cell, and secondly, given that some locations spatial locations had depths lower than 500m, why they did not standardise the index to a depth common to all the locations?

Author's Response to Decision Letter for (RSPB-2021-2613.R0)

See Appendix A.

Decision letter (RSPB-2021-2613.R1)

17-Feb-2022

Dear Dr Shelton

I am pleased to inform you that your manuscript entitled "Environmental DNA provides quantitative estimates of Pacific hake abundance and distribution in the open ocean." has been accepted for publication in Proceedings B.

Data Accessibility section

Open Access

Corresponding authors from member institutions (<http://royalsocietypublishing.org/site/librarians/allmembers.xhtml>) receive a 25% discount to these charges. For more information please visit <http://royalsocietypublishing.org/open-access>.

Your article has been estimated as being 9 pages long. Our Production Office will be able to confirm the exact length at proof stage.

Paper charges

Sincerely,

Professor Gary Carvalho

Associate Editor:

Comments to Author:

Thank you for your thorough responses to the reviewer criticisms and for the clear tracked changes in the LaTeX-generated PDF. It was a pleasure handling this manuscript.